High diversity and suggested endemicity of culturable Actinobacteria in an extremely oligotrophic desert oasis

Arocha-Garza Hector Fernando 1
Canales-Del Castillo Ricardo 2
Eguiarte Luis E. 3
Souza Valeria 3
De la Torre-Zavala Susana susana.delatorrezv@uanl.edu.mx 1
1 Facultad de Ciencias Biológicas, Instituto de Biotecnología, Universidad Autónoma de Nuevo León , San Nicolás de los Garza , Nuevo León , Mexico
2 Facultad de Ciencias Biológicas, Laboratorio de Biología de la Conservación, Universidad Autónoma de Nuevo León , San Nicolás de los Garza , Nuevo León , Mexico
3 Departamento de Ecología Evolutiva, Instituto de Ecología, Universidad Nacional Autónoma de México , Mexico City , Mexico
Hutchings Matt
Electronic publication date: 2017 May 2
Publication date: 2017
Volume: 5
Electronic Location ID: e3247
Received 2016 Dec 23; Accepted 2017 Mar 29
Copyright: ©2017 Arocha-Garza et al.
Copyright year: 2017
Copyright holder: Arocha-Garza et al.
License: This is an open access article distributed under the terms of the Creative Commons Attribution License, which permits unrestricted use, distribution, reproduction and adaptation in any medium and for any purpose provided that it is properly attributed. For attribution, the original author(s), title, publication source (PeerJ) and either DOI or URL of the article must be cited.
License URL: https://creativecommons.org/licenses/by/4.0/

Keywords: Actinobacteria, Cuatro Cienegas, Endemism, Diversity, Streptomyces

Funding: CONACYT 221963 Alianza WWF-Fundación Carlos Slim This research received funding from CONACYT (México) through Ciencias Básicas 2013 Project no. 221963 for Susana De la Torre Zavala. This research was also supported by Alianza WWF-Fundación Carlos Slim to Valeria Souza and Luis E. Eguiarte. There was no additional external funding received for this study. The funders had no role in study design, data collection and analysis, decision to publish, or preparation of the manuscript.

==============================
The phylum Actinobacteria constitutes one of the largest and anciently divergent phyla within the Bacteria domain. Actinobacterial diversity has been thoroughly researched in various environments due to its unique biotechnological potential. Such studies have focused mostly on soil communities, but more recently marine and extreme environments have also been explored, finding rare taxa and demonstrating dispersal limitation and biogeographic patterns for Streptomyces. To test the distribution of Actinobacteria populations on a small scale, we chose the extremely oligotrophic and biodiverse Cuatro Cienegas Basin (CCB), an endangered oasis in the Chihuahuan desert to assess the diversity and uniqueness of Actinobacteria in the Churince System with a culture-dependent approach over a period of three years, using nine selective media. The 16S rDNA of putative Actinobacteria were sequenced using both bacteria universal and phylum-specific primer pairs. Phylogenetic reconstructions were performed to analyze OTUs clustering and taxonomic identification of the isolates in an evolutionary context, using validated type species of Streptomyces from previously phylogenies as a reference. Rarefaction analysis for total Actinobacteria and for Streptomyces isolates were performed to estimate species’ richness in the intermediate lagoon (IL) in the oligotrophic Churince system. A total of 350 morphologically and nutritionally diverse isolates were successfully cultured and characterized as members of the Phylum Actinobacteria. A total of 105 from the total isolates were successfully subcultured, processed for DNA extraction and 16S-rDNA sequenced. All strains belong to the order Actinomycetales, encompassing 11 genera of Actinobacteria; the genus Streptomyces was found to be the most abundant taxa in all the media tested throughout the 3-year sampling period. Phylogenetic analysis of our isolates and another 667 reference strains of the family Streptomycetaceae shows that our isolation effort produced 38 unique OTUs in six new monophyletic clades. This high biodiversity and uniqueness of Actinobacteria in an extreme oligotrophic environment, which has previously been reported for its diversity and endemicity, is a suggestive sign of microbial biogeography of Actinobacteria and it also represents an invaluable source of biological material for future ecological and bioprospecting studies.

Introduction

The phylum Actinobacteria are gram-positive bacteria with a high G+C content, and it constitutes one of the largest phyla within the Bacteria domain (Parte et al., 2012). Actinobacteria diversity and community structure have been thoroughly researched in various environments. However, such studies had focused mostly in soil communities (Coombs & Franco, 2003; Gremion, Chatzinotas & Harms, 2003; Mohammadipanah & Wink, 2015; Zhao et al., 2016); but more recently, marine environments have also been explored (Ward & Bora, 2006; Maldonado et al., 2009; Claverias et al., 2015; Duran et al., 2015; Chen et al., 2016; Mahmoud & Kalendar, 2016; Undabarrena et al., 2016).

As an indicator of their ecological importance, Actinomycetes, filamentous members of the phylum Actinobacteria account for about 10% of bacteria colonizing marine aggregates (Grossart et al., 2004). Initially, marine Actinomycetes were poorly characterized (Goodfellow & Williams, 1983), but more recently, culture independent studies have shown that marine Actinomycetes are diverse and abundant (Ward & Bora, 2006). Rare marine Actinomycetes taxa have been isolated from a range of depths, sediments and other microbial communities such as stromatolites (Allen et al., 2009). Actinomycetes also comprise about 10% of the microbiome of extreme habitats, showing extensive taxonomic diversity (Kuhn et al., 2014; Mohammadipanah & Wink, 2015; Liu et al., 2016; Qin et al., 2016). However, careful population studies must still be done to determine if Actinomycetes are cosmopolitan, or if they do have local ecotypes, i.e., some degree of biogeography. Endemism would be the clearest demonstration of microbial biogeography, as it is for other organisms such as Salinispora (Jensen, Dwight & Fenical, 1991; Johnson, 2005; Jensen & Mafnas, 2006; Winsborough, Theriot & Czarnecki, 2009; Coghill et al., 2013; Prieto-Davo et al., 2013). Nevertheless, to unambiguously accept the idea of unlimited dispersal of microorganisms, we need data from studies employing good sampling. Such is the case, for example, of Escherichia coli, human-related strains of which travel with their host all around the world, or the case of Bacillus subtilis that can form endospores and travel with the air (Souza et al., 2012a; Souza et al., 2012b). Even in such cosmopolitan bacteria, there are local ecotypes that are unrelated to any other known strains (Gonzalez-Gonzalez et al., 2013; Avitia et al., 2014; Valdivia-Anistro et al., 2015). Streptomyces, a filament and spore producer, and the most extensively studied genera of Actinomycetes, has been studied and it had shown environmental gradients and regional endemism in some localities (Davelos et al., 2004; Antony-Babu, Stach & Goodfellow, 2008; Kinkel et al., 2014; Andam et al., 2016).

Actinobacterial diversity and community structure have been thoroughly investigated, not only for their ecological importance, but also by virtue of their unique biotechnological potential due to their robust secondary metabolism and incomparable ability to produce a plethora of bioactive molecules with extensive medical, industrial and agricultural applications. Actinomycetes, are the source of most clinically relevant antibiotics in use today (Barka et al., 2016). Nevertheless, the growing emergence of antibiotic multirresistant pathogenic strains, challenges the scientific community to overcome the problem of rediscovery of known compounds. Recent studies have concluded that discovery of unkown bioactive molecules will be facilitated by focusing heavily on “gifted” (secondary-metabolites-rich), readily culturable microbes that have been isolated from untapped environments, such as marine ecosystems, which enhance the isolation of large-genome (>8 Mb), thus, rare culturable bacteria (Tiwari & Gupta, 2012; Zotchev, 2012; Subramani & Aalbersberg, 2013; Tiwari & Gupta, 2013; Baltz, 2016; Katz & Baltz, 2016; Smanski, Schlatter & Kinkel, 2016).

Correspondingly, efforts towards describing the extent of the diversity of culturable actinomycetes on different conditions and extreme environments have been done, as evidenced by recent reports of bioprospecting and diversity studies of actinobacteria on deserts, marine sediments and vents, coral reefs, glaciers, as well as in symbiotic relationships (Maldonado et al., 2009; Rateb et al., 2011; Lee et al., 2014; Duncan et al., 2015a; Duran et al., 2015b; Jami et al., 2015; Kuang et al., 2015; Mohammadipanah & Wink, 2015; Trujillo et al., 2015; Yang et al., 2015; Andam et al., 2016; Chen et al., 2016; Liu et al., 2016; Mahmoud & Kalendar, 2016; Undabarrena et al., 2016).

To assess the extent of morphological and metabolic diversity and the distribution of culturable actinobacteria populations on a local scale, we chose the extremely oligotrophic and biodiverse Cuatro Cienegas Basin (CCB), an endangered oasis in the Chihuahuan desert (Souza et al., 2012a; Souza et al., 2012b). This is a site where endemic Bacillus (Alcaraz et al., 2008; Cerritos et al., 2011), Pseudomonas (Escalante et al., 2009) and Exiguobacterium (Rebollar et al., 2012) have been described. Particularly, within the CCB, the Churince System has been studied with more intensity by a large team of scientists since it is the most endangered hydrological system due to its relatively high altitude within the valley (730 m above sea level, compared to, ca. 700 m above sea level which is the average of most of the CCB), and because the San Marcos Sierra near this site of the basin is too step to efficiently recharge the aquifer locally. Hence, the system depends mostly on deep ancient water with a magmatic influence (Wolaver et al., 2012). This, together with the calcium sulfate soil matrix, and extreme oligotrophy in terms of phosphorus-limitation (Elser et al., 2005), makes Churince the most unusual site within the CCB (Minckley & Cole, 1968). This analysis is relevant not only for understanding the extensive biodiversity of this bacteria in such a peculiar environment, but also for allowing us the biological material for the elucidation of biochemical strategies for survival in conditions of scarcity, future experimentation of bioactive molecules, as well as studies of ecological interactions, including cooperation and competition analyses to understand the processes that are relevant to structure these complex bacterial communities. In contrast to what is commonly expected in an extremely oligotrophic site, we found high morphological and unique taxonomic diversity of culturable Actinobacteria, and we were able to isolate enriched abundance of the genus Streptomyces. When compared to available databases, we observed six novel monophyletic clades and seven single-member clusters, containing a total of 31 OTUs of the genus Streptomyces that are presumably different from other species previously described, and thus good candidates for consideration as endemic to the CCB. These unique groups of Streptomyces strains represent key clades in evolutionary history of an anciently divergent Phylum of the Bacteria domain.

Materials and Methods

Study site and sampling

The Churince hydrological system (Fig. 1) is located in the western part of the CCB, at 740 m above sea level, surrounded by large and mostly pure gypsum dunes. This system consists of three main zones connected by small water causeways: a spring, an Intermediate Lagoon (IL), and a desiccation lagoon (Lopez-Lozano et al., 2013). The Intermediate Lagoon (IL), where sampling took place, has low seasonal variations such as: salinity ranging ∼1.5–7.1 ppt, pH 7.6 to 8, and water temperature fluctuation from 14–20°C in winter and 20 to 30°C in summer (data of this study).

Figure 1 The Churince hydrological system.

(I) Map of Mexico displaying the State of Coahuila and the location of the Cuatro Cienegas Basin (CCB) and the Churince hydrological system (circle) © 2016 INEGI. (II) Aerial view of the intermediate lagoon (IL) in the Churince hydrological system. The circular forms point out the sampling sites. Image © 2016 DigitalGlobe © 2016 Google © 2016 INEGI.

Sampling took place during 2013–2016 at the following times: February 2013, March 2013, October 2013, October 2014, January 2015, February 2015, July 2015, April 2016. Samples were obtained from water and upper layer sediment from six locations along the shore in the Intermediate Lagoon in the Churince system (Fig. 1) in Cuatro Cienegas, Coahuila with the permission of Federal authorities to collect in the Natural Protected Area (SEMARNAT scientific sampling permit No. SGPA/DGVS/03121/15): Location A: 26∘50′53.79″N, 102°08′30.29″W; location B: 26°50′53.53″N, 102°08′31.81″W; location C: 26°50′54.37″N, 102°08′32.96″W; location D: 26°50′55.30″N, 102°08′33.63″W; location E: 26°50′55.63″N, 102°08′35.28″W; location F: 26°50′56.57″N, 102°08′36.03″W. At each site, water and surface sediments (0.2=-1 cm) were transferred to sterile conical tubes (50 ml). Samples were transported to a nearby laboratory in the town of Cuatro Cienegas at room temperature (≤1.5 h) and were used for streaking out primary plates immediately.

Selective isolation of culturable Actinobacteria

Nine selective Actinobacterial Isolation Media (AIM) were designed for this work to enhance the isolation of actinobacteria of aquatic and sediment environment. AIM1 ([per liter]: 21g yeast extract agar, 10g Malt extract, 4g Dextrose, 25g Reef salt mix); AIM2 ([per liter]: 20g mannitol, 20g soy flour, 20g Agar, 25g Reef salt mix); AIM3 ([per liter]: 50g chitin, 16g agar, 25g Reef salt mix); AIM4 ([per liter]: 10g starch, 1g Casein, 15g agar, 25g Reef salt mix); AIM5 ([per liter]: 20g Oat meal, 0.001 g Fe2(SO4)3, 0.001 g MgCl2, 0.001 g ZnSO4, 18g agar, 25g Reef salt mix); AIM6 [per liter]: 10g starch, 1g K2HPO4,1g H14MgO11S, 2g H8N2O4S, 1g NaCl, 2g CaCO3, 0.001 g FeH14O11S, 0.001 g MgCl2, 0.001 g ZnSO4, 20g agar, 25g Reef salt mix); AIM7 ([per liter]: 40g Soy Tripticasein agar, 25g Reef salt mix); AIM8 ([per liter]: 10g Bactopeptone, 5g Yeast extract, 16g agar, 25g Reef salt mix ); AIM9 ([per liter]: 100 µl humic acid, 0.02 g CaCO3, 0.5 g Na2HPO4, 0.5 g MgSO4, 1.7 g KCl, 0.01 g FeSO4, 0.5 mg Vitamin B12, 18g agar, 25g Reef salt mix).

All isolation media were autoclave-sterilized and supplemented with 0.20 µm pore size filtered Nystatin (100 µg/ml) to inhibit fungal growth, nalidixic acid (50 µg/ml) to inhibit gram-negative bacteria growth and to favor the growth of slow-growing Actinobacteria.

Prepared media were used for primary selective isolation of Actinobacteria by plating 150 µl directly from fresh samples, and using sterile 3 mm glass beads. Inoculated plates were incubated at 27°C for 1–6 weeks. Isolates were selected based on colony morphology and Gram stain, picked and re-streaked several times to obtain pure cultures. Isolates were maintained on AIM1 and AIM6 agar plates for short-term storage, and long-term strain collections were set up in 50% glycerol and preserved at −20°C (sporulated) and −80°C (non-sporulated).

Nucleic acid extraction

To confirm Actinobacteria identity and further phylogenetic analysis of isolates, after testing several techniques, genomic DNA was prepared using a modified phenol/ chloroform method that yielded the best quality DNA for our isolates: colonies of putative Actinobacteria were carefully scraped from agar plates and placed in centrifuge tubes; cell pellets were washed 2× 10 ml of 10% (w/v) with sucrose and resuspended in 400 µl of lysis solution (4% Triton x-100, 20% SDS, 5M NaCl, 2M Tris–HCl pH 8, 500mM EDTA pH 8). After resuspension, 400 µl of Phenol/Chloroform and 0.1 mm glass beads were added to lysis mix and this was mechanically disrupted for 2 min. The lysates were centrifuged (12,000 x rpm, 15 min) and DNA in aqueous phase was precipitated with 2 volumes of ethanol and 1/10 volume of 3M sodium acetate, pH 5.2; after overnight incubation at −20°C, DNA was centrifuged (12,000 x rpm, 10 min at 4°C), washed with 70% ethanol and eluted in TE with RNase.

Molecular identification and phylogenetic analysis

Genomic DNA from putative Actinobacteria was sent to Macrogen, Inc., USA, to perform 16S rDNA gene amplification by PCR and sequencing using the universal primers 27F (5′-GAGTTTGATCCTGGCTCAG-3′) and 1492R (5′-TACGGYTACCTTGTTACGACTT-3′), as well as phylum-specific primers: S-C-Act-235-a-S-20 (5′CGCGGCCTATCAGCTTG TTG-3′) (Stach et al., 2003) and 23SR (5′-AGGCATCCACCGTGCGCCCT3′) (Yoon et al., 1997).

The 16S rDNA gene sequences were edited and assembled using CodonCode Aligner 5.1 software (CodonCode Corporation, Dedham, MA); assembled contigs were compared to 16S rDNA gene sequences in the NCBI database (http: //www.ncbi.nlm.nih.gov/) using the Basic Local Alignment Search Tool (BLAST) to determine genus-level affiliations and are deposited in GenBank, which is associated with this document and are also available as Supplemental Information.

Our 16S rDNA gene sequences sharing a phylogenetic affiliation with Actinobacteria and reference sequences were aligned with ClustalW (Higgins, 1994) using Molecular Evolutionary Genetics Analysis MEGA Version 7 (Kumar, Stecher & Tamura, 2016).

Phylogenetic reconstructions were performed to analyze CCB OTUs clustering and taxonomic identification of the isolates in an evolutionary context. The phylogenetic tree of total Actinobacterial isolates was constructed by Maximum Likelihood (ML) algorithm using MEGA software v. 7 (Kumar, Stecher & Tamura, 2016) and Tamura–Nei I+G (Tamura, 1992) parameter as an evolutionary model with 1,000 replicates. For a more comprehensive interpretation of results, 16S sequences of previously characterized species of Actinobacteria with closest affiliations to our isolates, were obtained from GenBank databases and added to reconstructions of this Phylum. Criteria for selection of reference sequences was based on similarity and length of nucleotide sequences, but also, the selection of 16S sequences from study model organisms (such as S. coelicolor) and also microorganisms originally isolated from water and sediments from aquatic environments. Other reference strains were added to provide biological interpretation, and were selected from previous work reporting isolation of Streptomyces from deserts (Okoro et al., 2009; (Rateb et al., 2011)). Model selection was performed using statistical and evolutionary analysis of multiple sequence alignments TOPALi v2 (Milne et al., 2009).

Abundance and diversity were clearly remarkable for Streptomyces. From these early observations, we decided to compare distances between our Streptomyces isolates, to available information from previous studies, so we included a dataset of 667 16S-rDNA sequences of validated species of the Streptomycetaceae family; most of them were selected for a wide phylogenetic analysis within the family (Labeda et al., 2012 ; Labeda et al., 2017). We first performed a phylogenetic reconstruction using parameters and conditions reported by Labeda et al. (2012). Obtaining a preliminary Neighbour Joining (NJ) tree and leading us to the identification of relevant information regarding evolutionary relationships as well as the extent of the isolated diversity. It also provided criteria for selection of ideal reference strains for a later, more stringent analysis.

To reconstruct a second phylogenetic tree of the members of family Streptomycetaceae, we used the Maximum-likelihood (ML) method using MEGA software v. 7 and the Tamura–Nei I + G parameter as an evolutionary model. The reliability of nodes was estimated by ML bootstrap percentages (Felsenstein, 1985) obtained after 1,000 replications. A total of 41 16S sequences obtained in this study were included, and 73 reference strains belonging to the genera Streptomyces, 6 of Kitasatospora and 3 Streptoacidophilus, which were the most closely related to our isolates, were selected (trimmed to 1,074 bp).

To provide support to ML tree, we conducted a Bayesian analysis employing MrBayes v3.2.5 (Ronquist et al., 2012) with 10,000,000 Markov chain Monte Carlo generations and the GTR + G model of evolution with a nucmodel = 4by4, nruns = 2, nchains = 4, and sampled freq = 100. The average standard deviation of split frequences was below 0.001. The nodes that had posterior probabilities greater than 95 % (Bayesian), were considered well-supported and were shown in the resulting tree.

Estimation of diversity of Actinobacteria in CCB

To estimate species richness in the IL in the Churince system, we performed a rarefaction analysis for total Actinobacteria isolates, and another for only Streptomyces isolates. The definition of operational taxonomic units (OTUs) was conducted with MEGA software v. 7 at 97% cutoff according to their pairwise distances. Then we conducted the rarefaction curve using the EstimateS 9.1.0 software package (Colwell & Elsensohn, 2014) at the 95% confidence level.

Figure 2 (A) Pie chart of the percentage of Actinobacteria genera isolated from the intermediate lagoon in Churince system. (B) Number of Actinobacteria isolated according to the sampling sites. (C) Number of Actinobacterial isolated according to the culture media used.

Results

Diversity of culturable Actinobacteria within the Churince system in CCB

A total of 350 morphologically and nutritionally diverse isolates were successfully cultured and characterized as members of the Phylum Actinobacteria throughout the three-year period. AIM2 and AIM4 were the best nutrient conditions for culturing Actinomycetes (Fig. 2). Soy flour and mannitol-based medium allowed an isolation of five different genera of Actinobacteria and the greatest number of total isolates. The genus Streptomyces was found to be the most abundant taxa, accounting for over 50% of total sequenced isolates.

Diversity of cultured Actinobacteria varied in relation to sampling sites within the Churince. Among all sampling sites, C was the location where we found the highest diversity and abundance of Streptomyces strains. Only Streptomyces was ubiquitous in Churince IL and through the seasons, while isolation of the other 10 genera showed fluctuations.

From the entire isolated collection, 105 strains were successfully subcultured, processed for DNA extraction and 16S-rDNA sequenced (Table S1). These strains belong to the order Actinomycetales, and to suborders Corynebacterineae, Pseudonocardineae, Streptosporangineae, Frankineae, Streptomycineae, Micromonosporineae, Glycomycineae, and Micrococcineae, encompassing 11 genera of Actinobacteria. For phylogenetic analysis, a radial tree is presented in Fig. S1 showing the extent of macrodiversity of the genera of Actinobacteria retrieved from CCB.

Two rarefaction curves showed that the potentially yet-to-be-cultured diversity at both taxonomic levels (Actinobacteria phylum and Streptomyces genus) is large (Fig. 3); in fact, far higher than the 30 and 12 OTUs for Actinobacteria and Streptomyces, respectively, defined with a 97% cutoff according to their pairwise distances of the 16S-rDNA sequences, as seen by the curves, which are far from reaching the asymptote.

Figure 3 Rarefaction curves show sampling effort on the estimation of the numbers of OTUs at 97% sequence identity from cultured Actinobacteria (A), and total isolated Streptomyces (B) from CCB.

Figure 4 Colony morphological diversity of Streptomyces isolated from CCB within clades.

High diversity and phylogenetic clustering of Streptomyces from Cuatro Cienegas

Primary isolation plates were enriched with Streptomyces-like colonies in every sampling culture, with characteristic morphologies and geosmin-like odor. Streptomyces isolates account for 54% of the total sequenced isolates and since this genus was the most abundant in all media, sampling site and season, we first characterized these isolates based on their morphology to avoid picking clonal individuals for later DNA sequencing. Morphologies and other culture-related phenotypes varied among all selected individuals throughout the process of subculturing, such as colony morphology, pigment production, colony sporulation, optimal growth temperature and growth rate. Some of the different colony morphologies in Streptomyces are shown in Fig. 4.

A preliminary phylogenetic reconstruction of the family Streptomycetaceae was performed using isolates from this study and a dataset of 635 16S-rDNA sequences from Streptomyces previously used for a broad phylogenetic analysis within the family Streptomycetaceae (Labeda et al., 2012) (Fig. S2). The analysis shows that numerous CCB isolates are closer to each other and separated along the tree topology from most reference organisms. To construct a summarized and well-supported phylogenetic analyses, two different methods were used (Bayesian and ML), including 95 close reference strains, as well as sequences from isolates from the Atacama Desert and other ecologically similar isolates (Fig. 5). In this summarized analysis, we can unambiguously identify six novel monophyletic clades with 31 new OTUs and seven single-member clusters, all of them isolated in the present study.

Figure 5 Phylogenetic tree of Streptomycetaceae family based on nearly full-length 16s rRNA gene sequences and their closely related type strains based on the maximum likelihood (ML) method, constructed by Tamura–Nei I + G evolutionary model with 1,000 bootstrap replicates.

Bootstrap values for ML in the range from 0.7 to 1 were marked with black circles. Bayesian supports at nodes in ranges 0.95 to 1 were marked with a red triangles and Bootstrap values for neighbor-joining at ranges 0.6 to 1 in blue squares.

Discussion

Actinobacteria from oligotrophic CCB are diverse and abundant

Several different culture media were defined and applied for maximum recovery of culturable Actinobacteria in this study over a 3-year period, including different seasons. From this effort, 350 morphologically diverse isolates of Actinobacteria within the Churince system, were successfully cultured making a large, valuable, indigenous collection of different cultivated morphologies within one particular site. Nevertheless, due to well-known difficulties in genotyping this phylum (Yoon et al., 1997; Stach et al., 2003; Farris & Olson, 2007; Kumar et al., 2007), we were able to extract DNA and sequence 16S-rDNA of only 105 of them. In light of our observations of the abundance and uniqueness of the 16S sequence of the Streptomyces from the CCB and the reported biases from other studies in Actinobacteria (Hansen et al., 1998; Farris & Olson, 2007; Krogius-Kurikka et al., 2009; Rajendhran & Gunasekaran, 2011), it is not difficult to speculate that this group of microorganisms would require a different approach for a detailed characterization, such as whole-genome analysis of culturable strains. Ongoing work in our research group is applying this strategy for the most peculiar strains of our collection.

Although gram-positive bacteria are more commonly observed in organic rich habitats (Fenical, 1993), isolated strains from the extremely oligotrophic Churince IL encompass 11 genera of Actinobacteria (Fig. 2), which is comparable to the culturable diversity found in rich marine environments (Duncan et al., 2015; Duran et al., 2015; Kuang et al., 2015; Chen et al., 2016; Undabarrena et al., 2016). Interestingly, Streptomyces was the most abundant taxa, representing over 50% of the total sequenced isolates varying in relation to sampling point within the Churince system (Fig. 2). This result is comparable to the Streptomyces- enriched isolation in extreme environments such as the Atacama Desert (Okoro et al., 2009); nonetheless, CCB culturable diversity within the Phylum Actinobacteria is greater.

CCB culturable Streptomyces diversity is still far from being exhaustively explored as shown by rarefaction analysis (Fig. 3), suggesting a complex community structure, both in sediment and in the water column.

Morphological and genetic diversity of this phylum in the Churince does not come totally as a surprise since in concurrent studies using Illumina16S rRNA tags (V Souza et al., 2017, unpublished data) it was observed that Actinobacteria are the most successful lineage in CCB water, with the notable presence of genera Streptomyces, Yaniella, Arthrobacter, Trueperella, as well as several putative Actinobacteria from non-culturable marine lineages, in particular a strain closely related to the marine PeM15, which is very sensitive to nutrient enrichment (Lee et al., 2017, unpublished data) and other clades unique to soil and sediment. These analyses are consistent with our isolation efforts, which yielded abundant and diverse Streptomyces and abundant Arthrobacter isolates. It is possible to speculate that those several putative non-culturable Actinobacteria lineages detected by Illumina in concurrent projects, relate to our great numbers of cultured isolates which were not able to be detected by universal and phylum-specific primers.

Many interesting morphotypes could not be identified using 16S rDNA sequences; in addition, many were lost as the purification of a single colony proceeded. Success at bringing the environment into the laboratory culture is not sufficient for successful cultivability of bacteria. Subsequent culturing of Actinomycetes to obtain axenic (pure) cultures from the Churince, dramatically reduced the total number of unique pure isolates, suggesting obligate mutualism and cross-feeding (Tanaka et al., 2004; Kim et al., 2011; Seth & Taga, 2014).

It is quite interesting to observe that previous bacterial isolation efforts in the IL of the Churince in the CCB, using a culture-dependent approach initially based on thermo-resistant aquatic strains, did not lead to the isolation of Streptomyces individuals among the numerous isolated Actinobacteria (Cerritos et al., 2011). Many variables can play a role in this marked difference, most probably the different culture methods of Cerritos et al. (2011) through which thermoresistant bacteria in Marine Agar media were selected, thus enriching the isolation of Micrococcineae members. In contrast, our study applied several media with different carbon and nitrogen sources to maximize the possibility of culturing a wider diversity. Even so, the rarefaction curve shows that the potentially yet-to-be-cultured diversity is large (Fig. 3), as commonly occurs in highly diverse communities (Colwell, Mao & Chang, 2004; Colwell & Elsensohn, 2014).

Another possible factor that could explain differences between our study and Cerritos et al. (2011) is the years which passed between sampling periods, including possible temporal variation in the community structure. Notably in the CCB, after the time of the initial isolations described in Cerritos et al. (2011), a decline of the Churince aquifer occurred. As shown in experiments with UV and temperature increase in mesocosms (Pajares et al., 2013; Pajares, Souza & Eguiarte, 2015), endemic CCB Actinobacteria are particularly susceptible to perturbation. Hence, it is possible that enrichment of Streptomyces after 2010 is a succession response to the shrinkage and concomitant changes in the Churince aquifer system.

Endemicity of Streptomyces in CCB

As expected from previous studies finding endemic microorganisms at CCB (Alcaraz et al., 2008; Rebollar et al., 2012), we found 38 unique operational taxonomic units (OTU’s) for Streptomyces. Moreover, these 38 novel OTUs are in six new monophyletic clades in a deeply represented and well-supported phylogeny of the family Streptomycetaceae, which is a sign of endemicity. What makes this result unprecedented in a relatively very well-known cosmopolitan genus, Streptomyces (Barka et al., 2016), is the discovery of this degree of diversity and endemism in such an oligotrophic extreme environment.

Even though these data do not represent evidence of dispersal limitation per se, the phylogenetic clustering of OTUs of the CCB among themselves, and the genetic distance between OTUs from 667 reported species of Streptomycetaceae family from other sites around the world (Fig. 5 and Fig. S2), could be explained by migration limitation to and out of the CCB.

Relevance of culturing new Actinobacteria strains and lineages

Only a tiny fraction of the universal bacterial diversity has been pure cultured (Pace, 2009), and with this, the description of the biological diversity of the prokaryotic branch of the tree of life remains limited. Moreover, as culturable Actinobacteria diversity available for the study and characterization has been still insufficient when searching for bioactive compounds, there has been an increasing urge to culture untapped diversity within under-explored habitats (Katz & Baltz, 2016).

While genome mining represents a major paradigm shift for exploration of rare taxa (Cano-Prieto et al., 2015; Tang et al., 2015; Smanski, Schlatter & Kinkel, 2016), recent studies from genome mining for secondary metabolites gene clusters of unculturable Actinobacteria support the culturable approach for natural product discovery targeting “gifted microbes”, obtaining samples from unexplored habitats. In particular, untapped marine sediments are recommended when searching for cultivable potentially bioactive natural products from Actinobacteria (Baltz, 2016).

Although clades and clusters of CCB-isolates along the phylogeny might suggest that OTUs within the same groups are very close to each other, Fig. 4 shows distinctive morphologies that clearly reflect the uniqueness of each isolate. Hence, this collection of Actinobacteria from Cuatro Cienegas represents an invaluable source of great diversity for microbial ecology and biotechnology studies considering that: (i) phylogenies constructed with the sequenced portion of our collection indicate six novel clades of Streptomyces, but they only represent a third of the successfully cultured collection; (ii) this collection has been isolated from an environment of a diversity and endemicity, that has previously been considered comparable to that in the Galápagos Island (Souza et al., 2012a; Souza et al., 2012b), and as revealed by our six clades cointaining only CCB isolates (Fig. 5), it is quite likely that we have cultured several unique species yet to be described; (iii) the great diversity shown here has been calculated using the conserved 16S rDNA marker, but it is well known that single-gene phylogenies might not always reflect the evolutionary history of a species due to the high degree of horizontal gene transfer (Marri, Hao & Golding, 2006), a phenomenon particularly common in Streptomyces (Huguet-Tapia et al., 2016; Tian et al., 2016).

In conclusion, we can mention that our findings suggest a very high, albeit still uncalculated richness in microbial diversity in CCB, as well as suggested endemism. Our main result show that the CCB is not only a special place to study community structure where Actinobacteria diversity plays a major ecological role in such an oligotrophic environment, but it also represents a promising area for bioprospecting studies that will require concerted long-term efforts to search for genuine and substantial contributions to the discovery of natural products.

Supplemental Information

Table S1 Table containing data from Actinobacteria isolated from CCB

Sequence analysis of the cultured Actinobacteria isolated from the intermediate lagoon in Churince system, based on partial 16S rDNA gene sequencing.

Click here for additional data file.

Figure S1 Complete phylogenetic reconstruction of phylum Actinobacteria b ased on 16S rDNA sequence

Phylum Actinobacteria Phylogenetic tree based on 16S rRNA sequences using Maximum Likelihood (ML) algorithm for representative CCB-isolated Actinobacteria and their closely related type strains. Bootstrap values based on 1000 replicates.

Click here for additional data file.

Figure S2 NJ Phylogenetic tree for Complete Streptomycetaceae family

Phylogenetic tree based on a data set of 667 16S rRNA sequences of the Streptomycetaceae family using the neighbour-joining method. Bootstrap values based on 1000 replicates.

Click here for additional data file.

Data Set S1 Data set for phylogenies

Complete data set used for phylogenetic analyses.

Click here for additional data file.

We thank Hamlet Avilés Arnaut for his critical review of the manuscript and Gabriela Olmedo for her invaluable support and critical observations throughout the project. We also want to thank Mercedes Cortés for her assistance during microbiological work with the Streptomyces collection. We deeply acknowledge “Centro de Bachillerato Tecnológico Agropecuario #22” for providing facilities during the sampling period. Finally, we thank SEMARNAT for access to and permission to sample in the CCB Natural Protected Area.

Additional Information and Declarations

Competing Interests

Author Contributions

Field Study Permissions

DNA Deposition

Data Availability

Valeria Souza and Luis E. Eguiarte are Academic Editors for PeerJ.

Hector Fernando Arocha-Garza performed the experiments, analyzed the data, wrote the paper, prepared figures and/or tables.

Ricardo Canales-Del Castillo analyzed the data, contributed reagents/materials/analysis tools, prepared figures and/or tables, reviewed drafts of the paper.

Luis E. Eguiarte analyzed the data, contributed reagents/materials/analysis tools, wrote the paper, reviewed drafts of the paper.

Valeria Souza conceived and designed the experiments, analyzed the data, contributed reagents/materials/analysis tools, wrote the paper, reviewed drafts of the paper.

Susana De la Torre-Zavala conceived and designed the experiments, performed the experiments, analyzed the data, contributed reagents/materials/analysis tools, wrote the paper.

The following information was supplied relating to field study approvals (i.e., approving body and any reference numbers):

Access for sampling in Natural Protected Area in Cuatro Cienegas Coahuila was approved by Secretaría de Medio Ambiente y Recursos Naturales as stated by Permit: SGPA/DGVS/03121/15.

The following information was supplied regarding the deposition of DNA sequences:

The DNA sequences reported in these experiments have been uploaded as a Supplemental File.

The following information was supplied regarding data availability:

The raw data has been supplied as a Supplementary File.

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
