# Peer review of "High diversity and suggested endemicity of culturable Actinobacteria in an extremely oligotrophic desert oasis"

_PeerJ, doi:10.7717/peerj.3247_

## Round 0.1 · original submission · Major Revisions

As you will see from the comments under 'Validity of the findings', Reviewer 2 finds serious issue with the phylogenetic analysis. If you believe you can address these within the timeframe (5 days) then you are welcome to resubmit this manuscript but I will have to send it back to the same reviewer.

·

Basic reporting

The manuscript of Arocha-Garza describes a 3-year longitudinal study of the diversity of Actinobacteria from an extremely oligotrophic and biodiverse habitat in the Chihuahuan desert. The authors used a culture dependent approach using nine selective media and used universal and phylum specific 16S rDNA primers to examine the phylogenetic diversity of the isolates. The authors isolated 350 putative Actinobacteria and sequenced the 16S rDNA of 105 of these isolates. The work indicates that these isolates represent 23 novel isolates in 6 new monophyletic clades and highlight the actinobacterial diversity in this unique oligotrophic environment.

The manuscript is well written on the whole and makes an interesting contribution to the literature. The discussion is fairly long, and could potentially be shortened.

Experimental design

The Actinomycete isolation media used in the study have no reference associated with them, have these been used before? Can the authors provide some degree of validation for these media? This needs to be clear if they are novel media - also the use of AIM needs to be clearly defined to avoid ambiguity.

Validity of the findings

The authors could make more of the temporal data they have in terms of the isolations over the 3 year period. Can the authors see any differences in isolation rates?

Additional comments

Minor points

Abstract line 34 – ‘Putative Actinobacteria were sequenced using both bacteria universal and phylum-specific primers.’ This should read something like ‘the 16S rDNA of putative Actinobacteria were sequenced using both bacteria universal and phylum-specific primer pairs.’

Line 78-79 – the mention of biogeography may benefit from some references relating to the biogeography of Actinobacteria – work from the Jensen group on Salinispora may

I feel the authors should add some kind of description of oligotrophy to the introduction, moreover, it is not clear how/why this sample site was defined as oligotrophic, and perhaps the authors should include a reference for this, or some data in the results that enable the reader to understand the physiochemical environment that they have sampled from.

Line 266 – I would be careful of using ’geosmin odor’ and use ‘geosmin-like odour’ as other volatile chemicals such as methylborneol

·

Basic reporting

This manuscript reports on the characterisation of collection of actinomycetes from a Mexican desert environment.

The manuscript is generally well presented although minor edits and improvements to the English language usage are needed throughout. For example:
- Lines 26 and 58 “Actinobacteria” should be “Actinobacterial”
- Lines 28 and 60 “mostly in” should be “mostly on”
- Line 31 “populations in a” should be “populations at a”
- Line 83 “which human related strains do travel” should be “human-related strains of which travel”
- Line 86 “not identical to anything we know” is unclear: “unrelated to any other known strains”?
- Line 89 “the better studied” should be “most extensively studied”

I would recommend the authors have the manuscript checked by a native English speaker.

Figures need to be presented in order of appearance (Fig 3 is cited in methods before Fig 2).

The authors appear to use ESU and OTU interchangeably and so should standardise on one term.

Significant comment re literature cited and contextualisation: The discussion would be greatly improved by discussion of how the range of Actinobacteria recovered in the study compares to studies of the Actinobacteria typically recovered from other desert environments such as the Atacama (there are many such studies).

Experimental design

The Methods are presented in appropriate detail and the study is performed at a reasonable scale (350 initial isolates and 96 characterised), sampling an interesting location.

Laboratory methods seem appropriate.

However, it is noted that there is a rather rudimentary approach to the phylogenetic analysis (see 'validity of the findings').

Validity of the findings

The study would undoubtedly have been strengthened and could possibly have exposed a different story if it had been complemented by a culture-independent approach (hence the findings from the rarefaction curves lines 258-259). Consequently, the finding that members of the genus Streptomyces are the dominant taxon is hardly surprising (not for nothing is Streptomyces the largest known bacterial genus cf. line 89 of the manuscript).

In Figure 3 it is not clear why only 38 lineages from this study are shown in bold – these are not just the 23 novel OTUs so what ‘cut off’ applied to the selection of the lineages shown?

Line 49 claims there is lineage (ST_627_E) basal to the phylum Actinonacteria but it is clear from Line 288 that the authors mean basal to the genus Streptomyces. This ambiguity is further repeated at lines 366-367. Better evidence is needed that strain ST_627_E is basal to the genus, including construction of trees with a better selection of outgroups and representatives of genera closely related to Streptomyces in the family Streptomycetaceae, and also confirmation that the same topology is seen in trees constructed trees with additional methods other than the ML method. In this regard, the authors’ attention is drawn to a recent MLSA of the structure of the genus Streptomyces (Labeda et al. 2017, https://www.ncbi.nlm.nih.gov/pubmed/28039547).

Additional comments

It is a hard call as to whether this is a major revision or reject but on balance I think so much more work needs to be done on the phylogenetic analysis that revision within a reasonable period of time isn't feasible and I have had to recommend 'reject'.

---

## Round 0.2 · accepted · Accept

Many thanks for your detailed rebuttal and revised manuscript. I appreciate all your efforts and I think the manuscript is greatly improved as a result. I am therefore very happy to accept it for publication in PeerJ. Congratulations and all the best.